# Molecular evidence of widespread benzimidazole drug resistance in *Ancylostoma caninum* from domestic dogs throughout the USA and discovery of a novel β-tubulin benzimidazole resistance mutation

Abhinaya Venkatesan[1], Pablo D. Jimenez Castro[2,3,4], Arianna Morosetti[1], Hannah Horvath[1], Rebecca Chen[1], Elizabeth Redman[1], Kayla Dunn[2], James Bryant Collins[5], James S. Fraser[6], Erik C. Andersen[5], Ray M. Kaplan[2,7], John S. Gilleard[1]*

**1** Faculty of Veterinary Medicine, Host-Parasite Interactions Program, University of Calgary, Alberta, Canada, **2** Department of Infectious Diseases, College of Veterinary Medicine, University of Georgia, Athens, Georgia, United States of America, **3** Zoetis, Parsippany, New Jersey, United States of America, **4** Grupo de Parasitología Veterinaria, Universidad Nacional de Colombia, Colombia, **5** Molecular Biosciences, Northwestern University, Evanston, Illinois, United States of America, **6** Department of Bioengineering and Therapeutic Sciences, University of California, San Francisco, San Francisco, California, United States of America, **7** St. George's University, School of Veterinary Medicine, Grenada, West Indies

* jsgillea@ucalgary.ca

## Abstract

*Ancylostoma caninum* is an important zoonotic gastrointestinal nematode of dogs worldwide and a close relative of human hookworms. We recently reported that racing greyhound dogs in the USA are infected with *A. caninum* that are commonly resistant to multiple anthelmintics. Benzimidazole resistance in *A. caninum* in greyhounds was associated with a high frequency of the canonical F167Y(TTC>TAC) isotype-1 β-tubulin mutation. In this work, we show that benzimidazole resistance is remarkably widespread in *A. caninum* from domestic dogs across the USA. First, we identified and showed the functional significance of a novel benzimidazole isotype-1 β-tubulin resistance mutation, Q134H(CAA>CAT). Several benzimidazole resistant *A. caninum* isolates from greyhounds with a low frequency of the F167Y (TTC>TAC) mutation had a high frequency of a Q134H(CAA>CAT) mutation not previously reported from any eukaryotic pathogen in the field. Structural modeling predicted that the Q134 residue is directly involved in benzimidazole drug binding and that the 134H substitution would significantly reduce binding affinity. Introduction of the Q134H substitution into the *C. elegans* β-tubulin gene *ben-1*, by CRISPR-Cas9 editing, conferred similar levels of resistance as a *ben-1* null allele. Deep amplicon sequencing on *A. caninum* eggs from 685 hookworm positive pet dog fecal samples revealed that both mutations were widespread across the USA, with prevalences of 49.7% (overall mean frequency 54.0%) and 31.1% (overall mean frequency 16.4%) for F167Y(TTC>TAC) and Q134H(CAA>CAT), respectively. Canonical codon 198 and 200 benzimidazole resistance mutations were absent. The F167Y(TTC>TAC) mutation had a significantly higher prevalence and frequency in Western

**Data Availability Statement:** The near full-length isotype-1 β-tubulin gene sequences have been submitted to NCBI's GenBank and are available under the accession numbers: OP616973-OP616977. DNA amplicon sequencing data have been submitted to NCBI's Sequence Read Archive (SRA) and are available under the BioProject ID: PRJNA889845. Data on the high-throughput fitness assay used to measure the growth of C. elegans worms with the edited 134H ben-1 phenotype are available from https://github.com/AndersenLab/2022_BZ_Resistance_ben1_Q134H.

**Funding:** ECA, JSG, JSF, and RMK received funding support from the National Institutes of Health, USA (grant number R01AI153088). JSG also received funding from the Natural Sciences and Engineering Research Council of Canada (grant number 2021-02489) and the Bill and Melinda Gates Foundation (grant number OPP1172974). Additionally, PDJC was supported by funds from Boehringer Ingelheim, and AV and RC were supported by the Alberta Graduate Excellence Scholarship (AGES). The funders had no role in study design, data collection and analysis, decision to publish, or preparation of the manuscript.

**Competing interests:** The authors have declared that no competing interests exist.

USA than in other regions, which we hypothesize is due to differences in refugia. This work has important implications for companion animal parasite control and the potential emergence of drug resistance in human hookworms.

## Author summary

Although increasingly common in livestock, no reports of widespread anthelmintic resistance are confirmed in any companion animal or human gastrointestinal nematode parasite to date. The canine hookworm is a common intestinal zoonotic parasite of dogs with severe clinical impacts in young dogs, and for which control is dependent on regular anthelmintic use. We recently reported multiple anthelmintic drug resistance in *A. caninum* isolates from greyhounds derived from multiple locations in the USA likely caused by long standing intensive treatment regimens in kennels. In this study, we investigated benzimidazole resistance in *A. caninum* in pet dogs across the USA. We also identified and showed the functional significance of a novel benzimidazole isotype-1 β-tubulin resistance mutation in *A. caninum* from greyhounds that has not been previously reported in the field for any organism. We then determined that this novel mutation, as well as a previously characterized resistance mutation, were present, often at high frequency, in many *A. caninum* populations across the USA. This study reports the first evidence of widespread drug resistance for any parasitic nematode of companion animals and illustrates the power of molecular approaches to rapidly assess anthelmintic resistance in a region.

## Introduction

*Ancylostoma caninum* is one of the most important parasitic gastrointestinal nematodes of dogs worldwide, with severe infections causing anemia, and even death in younger puppies [1–4]. In addition, this parasite is also of public health relevance due to its zoonotic potential in causing cutaneous larval migrans in humans [5], and patent infections in humans have also been reported in tropical regions [6,7]. It is also an important model for human hookworm infection [8,9]. Treatment and control are dependent on the routine use of broad-spectrum anthelmintic drugs, the most commonly used in pet dogs in the USA being benzimidazoles (fenbendazole and febantel), tetrahydropyrimidines (pyrantel), and macrocyclic lactones (moxidectin and milbemycin). Although anthelmintic resistance has been widespread in gastrointestinal nematodes of grazing livestock for many years, it has generally not been considered a problem for gastrointestinal nematodes of domestic dogs [4]. Until recently, the only published reports of anthelmintic resistance in *A. caninum* were several cases of pyrantel resistance in dogs in Australia [10–12]. Historically, there has been little concern about the risk of resistance emerging in canine gastrointestinal parasites or the need for drug stewardship with respect to resistance mitigation.

In spite of the historical lack of concern regarding drug resistance, the management of *A. caninum* has been problematic in greyhounds in the USA for some time, with high prevalence and infection intensities. The high hookworm infection intensities in greyhounds are due, at least in part, to the housing conditions in kennels with large numbers of dogs and outdoor runs that favor hookworm transmission. This situation has led to an overreliance on frequent anthelmintic drug treatments to achieve hookworm control, leading to high drug selection pressure being applied to *A. caninum* specifically in greyhounds in racing and adoption

kennels [13]. A recent study confirmed that resistance to multiple anthelmintic drug classes is now common in hookworms in greyhounds originating from kennels of eight different states of the USA; *in vitro* assays confirmed that all *A. caninum* samples that were exposed to benzimidazoles and macrocyclic lactones were resistant [14].

Until recently, there has been less concern about anthelmintic resistance in *A. caninum* from pet dogs as they are subject to much less intense anthelmintic drug selection pressure than kenneled greyhounds. However, there is strong evidence of an increase in the prevalence of *A. caninum* in pet dogs: An analysis of over 39 million diagnostic reports in pet dogs reported an overall increase of 47% from 2012 to 2018 in hookworm prevalence from 2.02% in 2012 to 2.96% in 2018 [15]. Additionally, more than 3000 fecal samples collected from 288 off-leash dog parks across the USA in 2019, showed a prevalence of 7.1% for *A. caninum* [16]. Concurrent with this increase in prevalence, there appears to be a rise in cases reported by veterinarians of persistent *A. caninum* infections despite multiple rounds of treatment with different drugs. Further, a recent investigation of three canine hookworm cases in pet dogs that were refractory to treatment, showed compelling evidence of multiple anthelmintic drug resistance to benzimidazoles, macrocyclic lactones, and pyrantel using both *in vitro* and *in vivo* phenotypic assays [14,17,18], as well as *in vivo* efficacy based on a controlled clinical trial [19]. However, in contrast to greyhounds, no work has been done to investigate the prevalence of drug resistance in *A. caninum* from pet dogs in any country to date. Hence, we undertook this work to investigate the prevalence and distribution of anthelmintic resistance in hookworms in the USA pet dog populations.

Our current understanding of the molecular genetics of benzimidazole resistance in the strongylid nematode group enables a molecular approach to be used to investigate the prevalence and distribution of resistance for this particular drug class. Molecular screening for drug resistant parasites offers a number of advantages over both *in vitro* and *in vivo* phenotypic assays. This allows for larger-scale studies, is more amenable to standardization, and provides information about the molecular epidemiology of resistance. Recent genome-wide studies have confirmed that the single most important benzimidazole resistance locus in the *Haemonchus contortus* genome is the isotype-1 β-tubulin gene [20,21]. A variety of studies have shown that a number of different mutations at codons 167, 198, and 200 of this gene have been shown to be associated with benzimidazole resistance in *H. contortus* and a number of related nematode species of livestock. The frequency and distribution of these mutations—F167Y (TTC>TAC), E198A(GAA>GCA), E198L(GAA>TTA), E198V(GAA>GTA), and F200Y (TTC>TAC)—vary with the nematode species and geographical region [22–28]. Further, these mutations have all been shown to be sufficient to confer similar levels of benzimidazole resistance in *C. elegans* [29–31]. Additionally, several other amino acid substitutions in the *C. elegans ben-1* gene (Q131L, S145F, A185P, M257I, D404N, G104S, G142E, G142R, E198K, and R241H), not yet identified in parasites, have been shown to confer benzimidazole resistance [29,32]. Mutagenesis studies in fungi have also identified mutations conferring benomyl resistance at codons H6Y, Y50C, Q134L, A165V, E198K, F200Y, and M257 [33–37]. The isotype-1 β-tubulin F167Y(TTC>TAC) mutation, but not mutations at codons 198 and 200, was found to be present at high frequency in all three confirmed multi-resistant *A. caninum* isolates reported from pet dogs but not in susceptible isolates [17]. The importance of this mutation was further confirmed by a significant correlation between the egg hatch assay $IC_{95}$ and the F167Y(TTC>TAC) mutation in the greyhounds [14,17].

Here, we use a molecular genetic approach to investigate benzimidazole resistance in *A. caninum* isolated from pet dogs across the USA. We first report the discovery of a novel benzimidazole resistance mutation at codon 134 in the isotype-1 β-tubulin gene in *A. caninum* that has not been previously reported in any parasite or fungal species in the field. Genotype-

phenotype relationships from greyhounds suggest that benzimidazole resistance can be accounted for by the presence of either the F167Y(TTC>TAC) or the Q134H(CAA>CAT) mutations. Predictive structural modeling and CRISPR-Cas9 editing demonstrate the functional significance of this novel Q134H mutation. Deep amplicon sequencing showed that both the F167Y(TTC>TAC) and Q134H(CAA>CAT) mutations are widespread in *A. caninum* from pet dogs across the USA, demonstrating benzimidazole resistance is well established in *A. caninum* from pet dogs across the USA, which has important implications for the management of hookworms in dogs and drug stewardship. In addition, the results provide important information for the development and application of molecular diagnostics for benzimidazole resistance and also provide an important precedent for the emergence of resistance in other hookworm species including those of humans. Finally, this work illustrates the value of combining work in parasites with *C. elegans* for the discovery and confirmation of novel anthelmintic resistance mutations [38].

## Results

### Discovery of a novel benzimidazole resistance mutation in nematodes: isotype-1 β-tubulin Q134H(CAA>CAT)

Egg Hatch Assays (EHA) were previously conducted on *A. caninum* eggs isolated from a set of fecal samples from USA greyhounds, and all showed the presence of benzimidazole resistance [14] (Fig 1A). The canonical isotype-1 β-tubulin F167Y(TTC>TAC) benzimidazole resistance mutation was present at a high frequency in 13 out of the 16 samples (82.6%, range, 40.2% - 98.1%) but was absent in one sample and was present at low frequency in the remaining two samples (5.7%, 5.3%) (Fig 1B). The other canonical isotype-1 β-tubulin benzimidazole resistance mutations at codons 198 and 200 were absent from all 16 *A. caninum* samples from greyhounds, suggesting the potential role of a novel mutation in conferring benzimidazole resistance phenotypes in these three samples. Examination of the Amplicon Sequence Variants (ASVs) of the 293 bp isotype-1 β-tubulin fragment previously generated by deep amplicon sequencing revealed a single non-synonymous mutation Q134H(CAA>CAT) in many of the greyhound *A. caninum* samples and at a high frequency (90.0%, 94.4%, and 100%) in the three samples in which the F167Y(TTC>TAC) was absent or at low frequency (Fig 1C). In order to determine if any additional non-synonymous mutations occurred elsewhere in the isotype-1 β-tubulin gene for the haplotype carrying the Q134H(CAA>CAT) mutation, we PCR-amplified and cloned a 3,467 bp fragment encompassing the near full-length isotype-1 β-tubulin gene from two of the greyhound *A.caninum* egg samples with a high frequency of this mutation as well as from the benzimidazole-susceptible sample (Barrow) as a control. Sequencing of six clones from the three samples revealed no additional non-synonymous mutations in the isotype-1 β-tubulin (S1 Fig). Consequently, we hypothesized the Q134H(CAA>CAT) to be a novel mutation responsible for the benzimidazole resistance phenotype in the three samples with a low frequency of the F167Y(TTC>TAC) mutation.

We undertook predictive structural modeling of the *A. caninum* isotype-1 β-tubulin polypeptide to assess the potential impact of the Q134H(CAA>CAT) mutation on benzimidazole drug binding (Figs 2 and S2). The amino acid positions associated with benzimidazole resistance—134, 167, 198, and 200—were mapped to the nocodazole (with thiabendazole modeled) binding pocket of β-tubulin and all of these sites, including the newly discovered Q134(CAA), are in direct Van der Waals contact with the methyl ester terminus of nocodazole. Clinically relevant molecules from the same drug class, such as albendazole, mebendazole, and fenbendazole, vary only at the opposite end of the molecule. Disruption of these contacts by mutation is

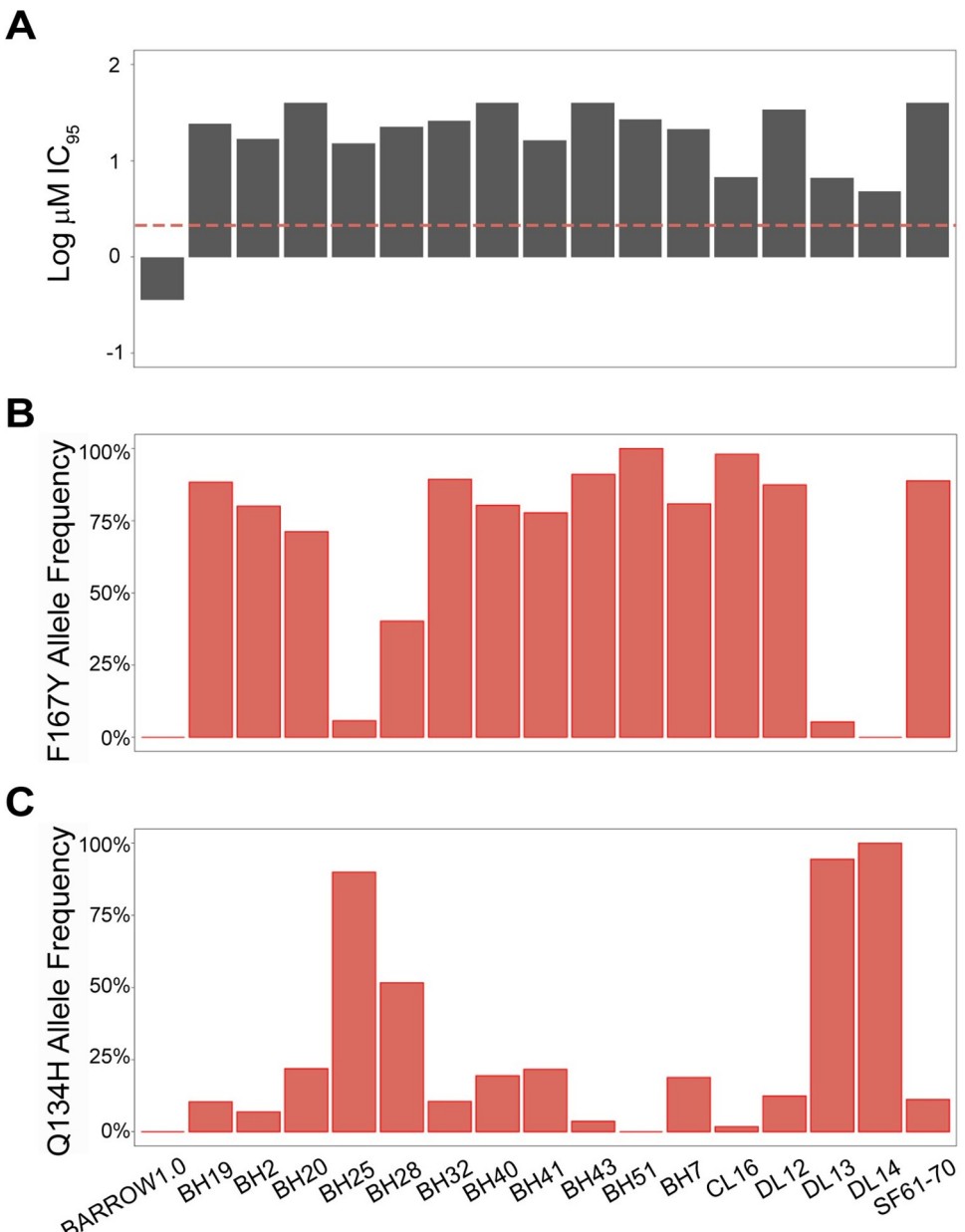

**Fig 1. Egg hatch assay and isotype-1 β-tubulin 167Y(T$\underline{A}$C) and 134H(CA$\underline{T}$) mutation frequency data for 16 *A. caninum* samples from 25 greyhounds.** (A): Log-transformed Egg Hatch Assay (EHA) IC$_{95}$ values (μM) for the 16 *A. caninum* fecal egg samples. The resistance threshold is indicated by the red dashed line; (B): 167Y(T$\underline{A}$C) isotype-1 β-tubulin mutation frequencies; (C): 134H(CA$\underline{T}$) isotype-1 β-tubulin mutation frequencies. Sample SF61-70 is a pool comprising *A. caninum* from 10 greyhounds. Barrow 1.0 is the first passage of an *A. caninum* benzimidazole-susceptible laboratory sample [10]. The remaining samples are from 25 greyhounds described in [12].

predicted to introduce steric clashes (Q134H, F167Y, F200Y) or lead to the loss of favorable interactions with the azole ring (E198L/A/V) (Fig 2).

In order to investigate whether the Q134H(CA$\underline{A}$>CA$\underline{T}$) mutation was sufficient to confer phenotypic resistance *in vivo*, we performed CRISPR-Cas9 genome editing to introduce a 134H(CA$\underline{C}$) allele into the *C. elegans ben-1* gene in the N2 genetic background. Two independently edited strains, *ean243* and *ean244*, were created to control for any off-target effects

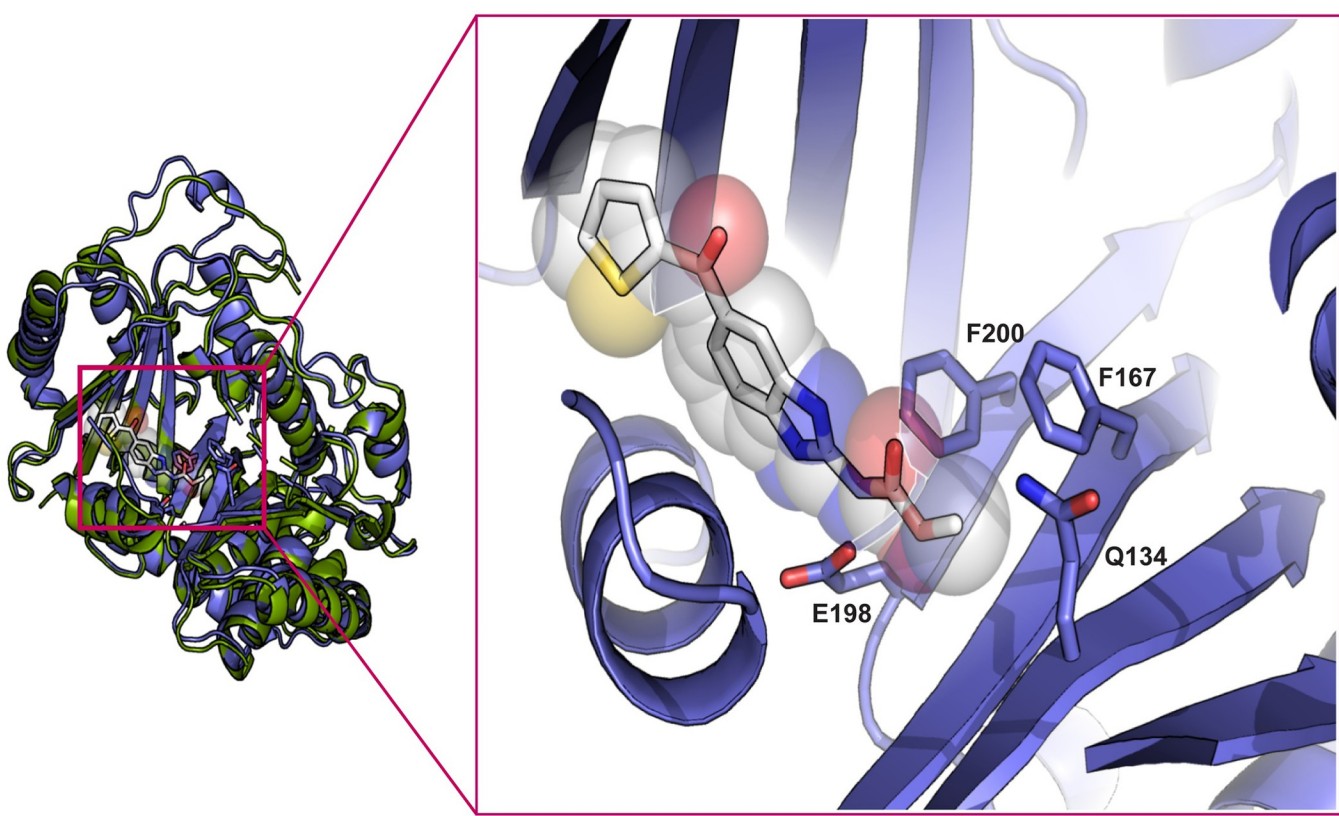

**Fig 2. *In-silico* protein structural model of the *A. caninum* isotype-1 β-tubulin bound to nocodazole.** Structural model for *A. caninum* isotype-1 β-tubulin made using AlphaFold2 (PMID: 34265844) (purple) aligned to Porcine β-tubulin (green) bound to nocodazole (white) (from PDB 5CA1). The inset shows a zoom into the binding site, with positions of residues at codons 134, 167, 198, and 200 indicated. These resistance mutation positions are all in direct van der Waals contact with the constant portion of benzimidazole drugs.

from genome editing. We measured the responses to albendazole exposure of the two *ben-1* 134H(CA<u>C</u>) genome-edited strains, as well as the N2 (susceptible control) and *ben-1* deletion (resistant control) strains (as well as DMSO as vehicle control). The 134H(CA<u>C</u>) *ben-1* allele showed significantly increased levels of resistance to albendazole relative to the susceptible N2 strain and similar levels of resistance as the *ben-1* deletion strain *(ean64)* (Fig 3). These results show that the Q134H(CA<u>C</u>) allele confers similar levels of resistance in *C. elegans* as a *ben-1* null allele, the same level as has been previously shown for the other *ben-1* resistance mutations, including the F167Y(T<u>T</u>C>T<u>A</u>C) mutation [30,31].

### The canonical F167Y(T<u>T</u>C>T<u>A</u>C) isotype-1 β-tubulin benzimidazole resistance mutation is widespread and often at a high frequency in *A. caninum* from pet dogs across the USA

The F167Y(T<u>T</u>C>T<u>A</u>C) isotype-1 β-tubulin benzimidazole resistance mutation has previously been shown to be common in *A. caninum* in greyhounds from racing and adoption kennels in the USA [14]. In order to investigate the situation in pet dogs, hookworm eggs were collected from individual pet dog fecal samples across the USA previously identified as hookworm positive by microscopy (kindly supplied by IDEXX Laboratories). Genomic DNA preparations were made from the hookworm eggs for 328 individual dogs for which a minimum of 300 hookworm eggs were recovered (range 300–14,000). In order to also examine dogs with low

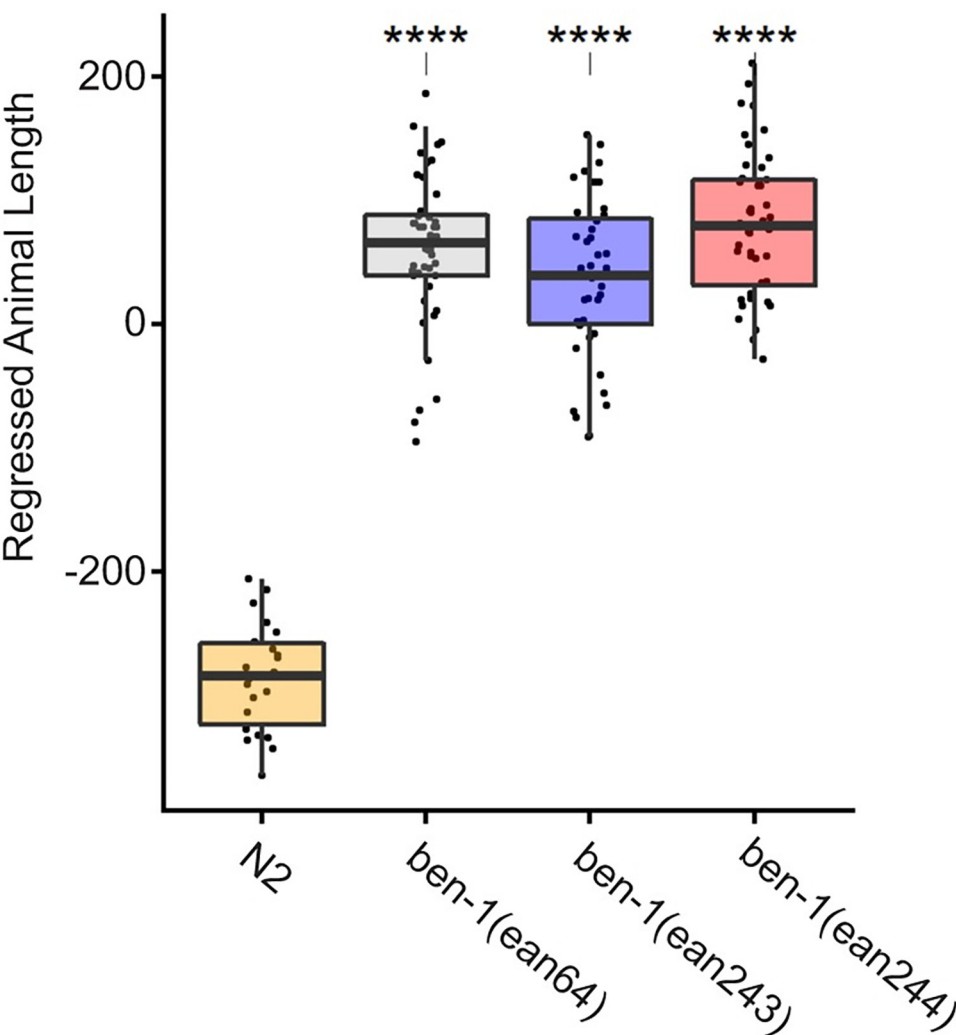

**Fig 3. Drug response assays for *ben-1* Q134H(CA$\underline{A}$>CA$\underline{C}$) alleles introduced into *C. elegans* by CRISPR-Cas9 editing.** Regressed median optical density (median.EXT), a proxy measurement of animal length, from populations of nematodes after exposure to 30 μM albendazole is shown. *ben-1(ean64)* is a previously created deletion strain. *ben-1 (ean243)* and *ben-1(ean244)* are two independent CRISPR-Cas9 edited Q134H(CA$\underline{A}$>CA$\underline{C}$) alleles. Each point represents the regressed phenotype of a well containing approximately 50 animals. Data are shown as Tukey box plots with the median as a solid horizontal line, and the top and bottom of the box representing the 75th and 25th quartiles, respectively. The top whisker is extended to the maximum point within the 1.5 interquartile range from the 75th quartile. The bottom whisker is extended to the minimum point within the 1.5 interquartile range from the 25th quartile. Statistical significance is shown as asterisk above each strain ($p < 0.001 = $ ***, $p < 0.0001 = $ ****, Tukey HSD).

infection intensities (fewer than 300 eggs recovered), DNA lysates were prepared from 65 hookworm egg pools comprising 357 dogs, containing a minimum of 200 eggs per pool (range 200–10,000) (S1 Appendix and S1 Table). Deep sequencing was performed on two PCR amplicons generated from these DNA preparations—a 293 bp fragment encompassing codons 134 and 167, and a 340 bp fragment encompassing codons 198 and 200—and variant calling was used to determine the frequency of resistance mutations at each of these codons (Fig 4).

The amplicon encompassing codons 134 and 167 was successfully PCR amplified and deep sequenced from 314 individual samples (average read depth of ~19,930 reads; range: 1,028–95,636 reads). Variant calling at codon 167 revealed the F167Y(T$\underline{T}$C>T$\underline{A}$C) mutation was present in 49.7% (156/ 314) of the individual samples. The overall mean frequency in the

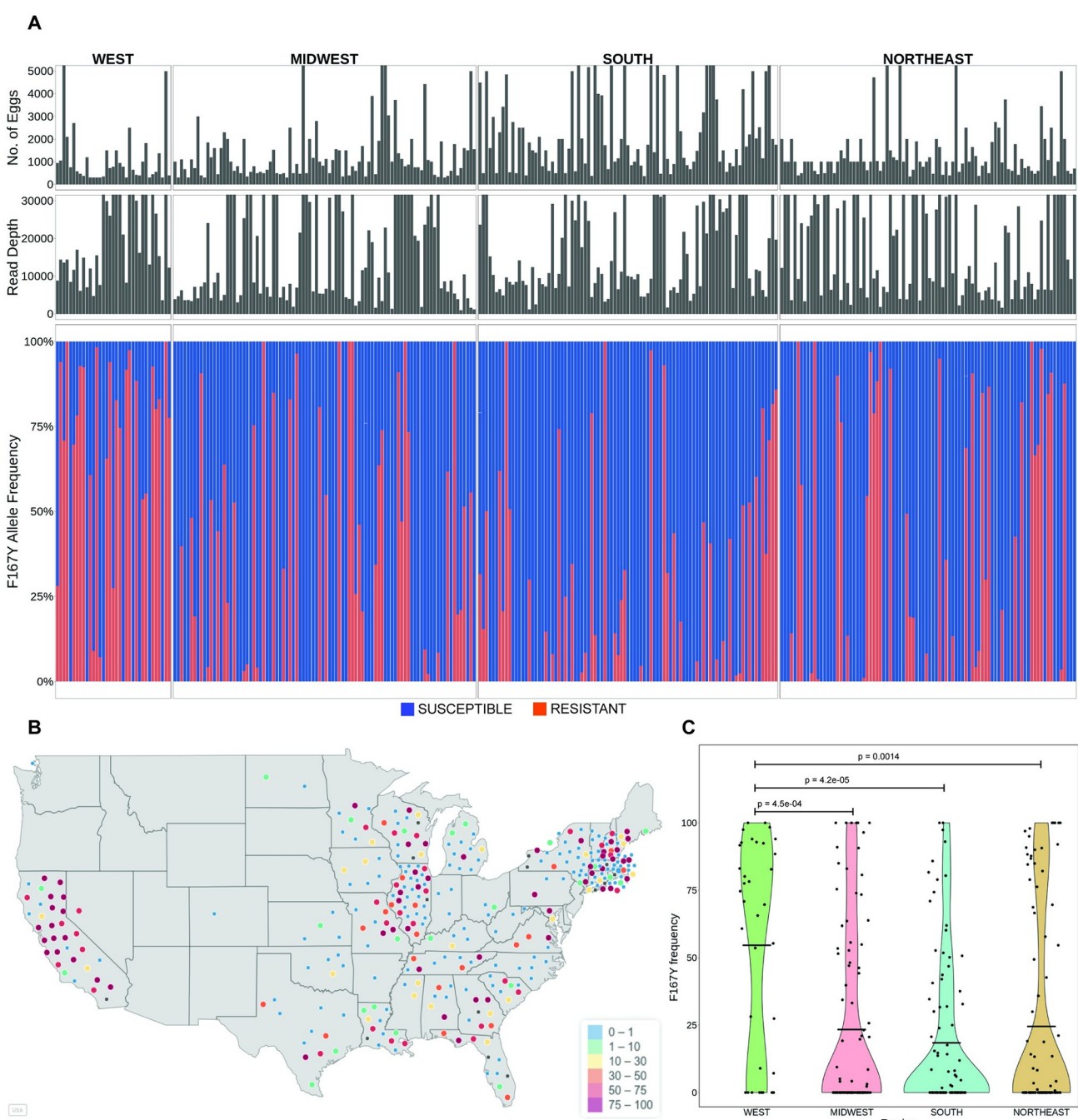

**Fig 4. Prevalence and frequency of the F167Y(TTC>TAC) isotype-1 β-tubulin benzimidazole resistance mutation in *A. caninum* from pet dogs across the USA.** (A): Deep amplicon sequencing data of the 293 bp isotype-1 β-tubulin fragment of *A. caninum* from the 314/ 328 individual samples across the USA. The top panel is a bar chart showing the number of eggs used to make each genomic DNA preparation, in the middle is a bar chart showing the mapped read depth for each sample, and the lower chart shows the relative frequency of the F167Y(TTC>TAC) mutation. Red bars indicate the 167Y(TAC) resistance allele and the blue bars indicate the 167F(TTC) susceptible allele. (B): Geographical distribution of the individual samples and the frequency of the 167Y(TAC) resistance allele. The map of the USA was generated using the *maps()* package in R and is available from: https://rdrr.io/ cran/maps/man/usa.html. The color of each circle indicates the frequency of the resistance mutation in each sample, ranging from 0–100%. (C): Violin Plot of the 167Y(TAC) resistance allele frequencies from samples in each of the four geographical regions. The mean frequency is indicated by a horizontal line and any statistically significant differences calculated using the pairwise Wilcoxon rank sum test (p<0.05) between the regions are indicated (p-value >0.05 not indicated).

**Table 1. Prevalence of the F167Y(TTC>TAC) mutation.**

| A: MEAN PREVALENCE OF F167Y ALLELE REGION WISE | | | | |
|---|---|---|---|---|
| Region | Samples sequenced | Samples carrying 167Y allele | Mean allele frequency | Bootstrap 95% CI |
| West | 35 | 27 | 72.8 | 62.4, 82.7 |
| Midwest | 94 | 44 | 52.7 | 42.9, 62.2 |
| South | 92 | 44 | 40.2 | 30.9, 48.6 |
| Northeast | 93 | 42 | 57.8 | 46.6, 68.3 |
| B: MEAN PREVALENCE OF F167Y ALLELE BY BREED SIZE | | | | |
| Breed size | Samples sequenced | Samples carrying 167Y allele | Mean allele frequency | Bootstrap 95% CI |
| Small | 50 | 18 | 45.5 | 29.2, 61.7 |
| Medium | 150 | 68 | 48.4 | 40.3, 56.3 |
| Large | 80 | 48 | 65.8 | 56.2, 74.7 |
| C: MEAN PREVALENCE OF F167Y ALLELE BY AGE OF THE DOG | | | | |
| Age category | Samples sequenced | Samples carrying 167Y allele | Mean allele frequency | Bootstrap 95% CI |
| Puppies (A) | 61 | 36 | 49.7 | 39.8, 60.5 |
| Young Adults (B) | 85 | 42 | 50.3 | 39.3, 61.5 |
| Mature Adults (C) | 32 | 12 | 65.1 | 41.7, 84.5 |
| Seniors (D) | 28 | 7 | 63.6 | 34.7, 88.6 |

positive samples was 54.0% (Bootstrap 95% C.I. 48.7% - 59.2%) and a frequency of >50% was present in 87 of these samples (Fig 4A and 4B).

There was a significant difference in the frequency of the 167Y(TAC) resistance mutation across the regions (Kruskal-Wallis, p = 1e-04). The prevalence and frequency of the F167Y (TTC>TAC) mutation were statistically significantly higher in the west than in the other three regions of the USA (75% prevalence and 72.8% overall mean frequency in positive samples from the west) (Fig 4D and Table 1). The frequency of this mutation was also significantly higher in large breeds of dogs compared to small and medium-sized breeds based on the data from 280 dogs for which breed information was available (Kruskal-Wallis, p = 0.002) (S3A Fig). Out of the 206 dogs with age-related information, the prevalence of the 167Y(TAC) mutation was significantly higher in puppies when compared to seniors (Fisher's test, p = 0.02), but there were no other differences in the frequency of the mutation among the different age groups (S3B Fig).

The amplicon encompassing codons 198 and 200 was successfully PCR amplified and deep sequenced from 312 individual samples (average read depth 23,084 reads; range: 1,039–247,226 reads). Variant calling at codons 198 and 200 revealed that 100% of these samples contained susceptible alleles at these codons (S4 Fig).

## The novel Q134H(CAA>CAT) isotype-1 β-tubulin benzimidazole resistance mutation is widespread in *A. caninum* from both pet dogs and greyhounds across the USA

The prevalence and distribution of the novel 134H(CAT) isotype-1 β-tubulin benzimidazole resistance mutation in pet dogs across the USA was also investigated by variant calling on the 293 bp amplicon sequencing data (314 individual samples with an average read depth of ~19,930 reads, range, 1,028–5,636 reads). The Q134H(CAA>CAT) mutation was detected in 31.2% (98/ 314) of the individual samples. Its overall mean frequency in these positive samples was 16.4% (Bootstrap 95% C.I. 13.0% - 20.1%), and it was at a frequency of >50% in five samples (Fig 5A and 5B).

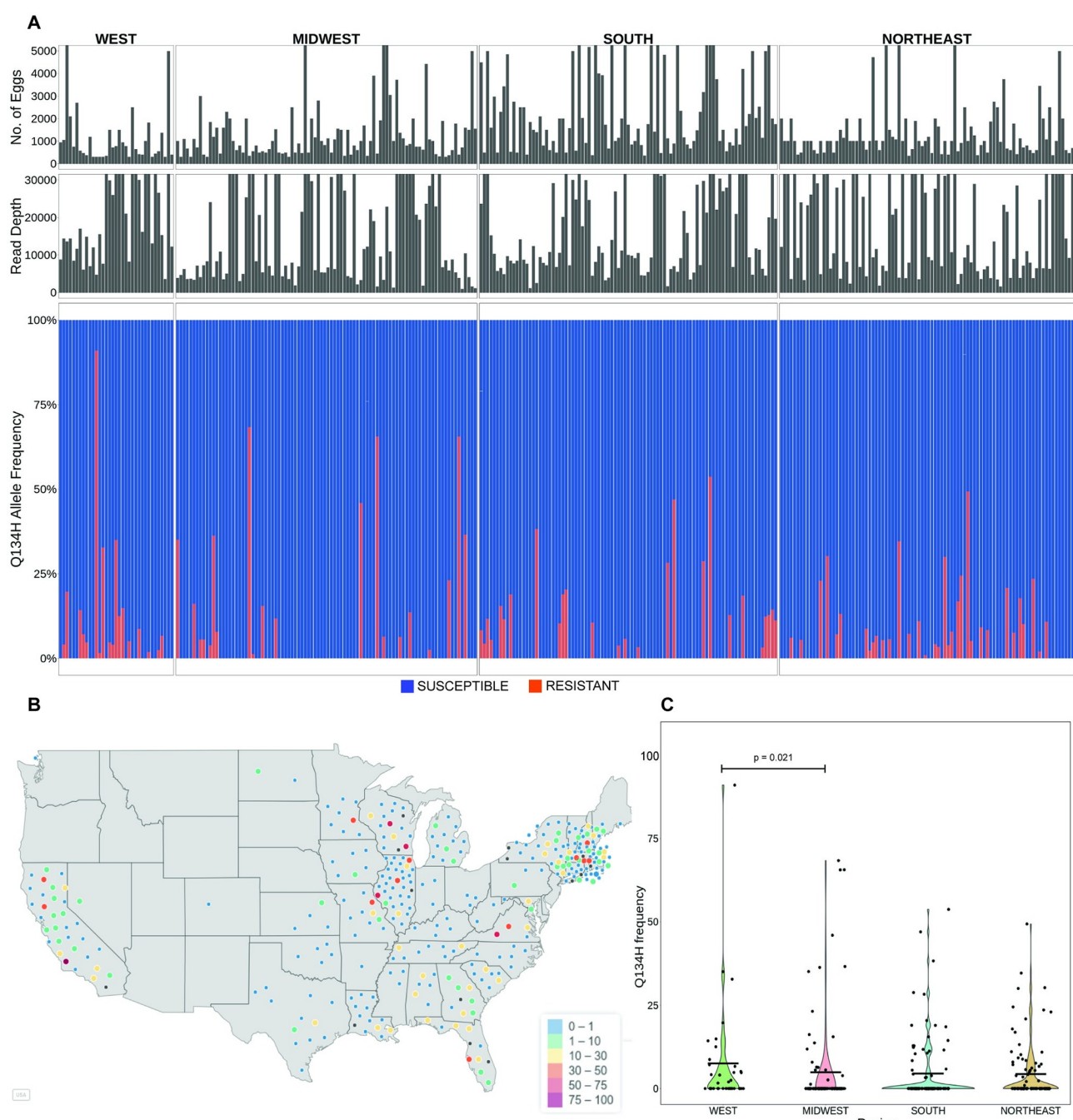

**Fig 5. Frequency and prevalence of the novel Q134H(CAA>CAT) benzimidazole resistance mutation in *A. caninum* from pet dogs across the USA.** (A): Deep amplicon sequencing data of the 293 bp isotype-1 β-tubulin fragment of *A. caninum* from the 314/ 328 individual samples across the USA. The top panel is a bar chart showing the number of eggs used to make each genomic DNA preparation, in the middle is a bar chart showing the mapped read depth for each sample, and the lower chart shows the relative frequency of the Q134H(CAA>CAT) mutation. Red bars indicate the 134H (CAT) resistance allele and the blue bars indicate the 134Q(CAA) susceptible allele. (B): Geographical distribution of the individual samples and the frequency of the 134H(CAT) resistance allele. The map of the USA was generated using the *maps()* package in R and is available from: https://rdrr.io/cran/maps/man/usa.html. The color of each circle indicates the frequency of the resistance mutation in each sample, ranging from 0–100%. (C): Violin Plot of the 134H(CAT) resistance allele frequencies from samples in each of the four geographical regions. The mean frequency is indicated by a horizontal line and any statistically significant differences calculated using the pairwise Wilcoxon rank sum test (p<0.05) between the regions are indicated (p-value >0.05 not indicated).

**Table 2. Prevalence of the Q134H(CAA>CAT) mutation.**

| A: MEAN PREVALENCE OF Q134H ALLELE REGION WISE | | | | |
|---|---|---|---|---|
| Region | Samples sequenced | Samples carrying 134H allele | Mean allele frequency | Bootstrap 95% CI |
| West | 35 | 18 | 15.1 | 7.4, 25.4 |
| Midwest | 94 | 20 | 23.7 | 15.2, 33.4 |
| South | 92 | 26 | 16.6 | 11.8, 21.7 |
| Northeast | 93 | 34 | 12.6 | 9.2, 16.8 |
| B:MEAN PREVALENCE OF Q134H ALLELE BY BREED SIZE | | | | |
| Breed size | Samples Sequenced | Samples carrying 134H allele | Mean allele frequency | Bootstrap 95% CI |
| Small | 50 | 12 | 12.3 | 6.7, 19.1 |
| Medium | 150 | 36 | 17.6 | 12.7, 23.5 |
| Large | 80 | 38 | 11.0 | 8.7, 13.9 |
| C: MEAN PREVALENCE OF Q134H ALLELE BY AGE OF THE DOG | | | | |
| Age category | Samples sequenced | Samples carrying 134H allele | Mean allele frequency | Bootstrap 95% CI |
| Puppies (A) | 61 | 20 | 18.7 | 10.5, 28.0 |
| Young Adults (B) | 85 | 27 | 14.9 | 9.7, 20.7 |
| Mature Adults (C) | 32 | 13 | 12.6 | 7.8, 18.1 |
| Seniors (D) | 28 | 5 | 9.3 | 2.8, 20.2 |

The prevalence and frequency of the Q134H(CAA>CAT) resistance mutation were statistically significantly higher in the west compared to the midwest but not compared to the other two regions (overall comparison, Kruskal-Wallis, p = 0.031) (Fig 5C and Table 2). Based on the 280 and 206 samples with available breed and age information respectively, the Q134H (CAA>CAT) resistance mutation frequency was significantly higher in large dogs compared to small and medium-sized dog breeds (Kruskal-Wallis, p = 0.002) (S5A Fig), but there were no significant differences among the different age categories (S5B Fig).

We also examined previously generated amplicon sequencing data from *A. caninum* from greyhounds sourced from southern USA adoption and racing kennels for the presence of the Q134H(CAA>CAT) resistance mutation [14]. Variant calling at codon 134 on these data revealed that 88.6% (62/ 70) of the *A. caninum* samples from the greyhounds carried the novel Q134H(CAA>CAT) resistance mutation. Its overall mean frequency in these positive samples was 25.8% (Bootstrap 95% C.I. 20.1% - 31.0%) and it was at a frequency of >50% in eight dogs (Fig 6).

Only one other non-synonymous mutation was detected in the whole data set: a D128N (GAC>AAC) substitution that was detected at low frequency in just two out of the 70 kenneled greyhound samples (frequency 13.5% in both samples) but was not detected in any of the 314 pet dog samples. This mutation has not been described as associated with resistance in fungi or other nematodes and was not further investigated at this point due to its low occurrence and frequency.

## Discussion

### Implications of widespread benzimidazole resistance in *A. caninum* in pet dogs across the USA

Prior to this study, there was no information on the extent of anthelmintic resistance in *A. caninum* infecting the general domestic dog population of any country. Phenotypic assays such as the *in vivo* fecal egg count reduction test or the *in vitro* egg hatch assay and larval development tests are not suitable for large-scale studies. However, the identification of two

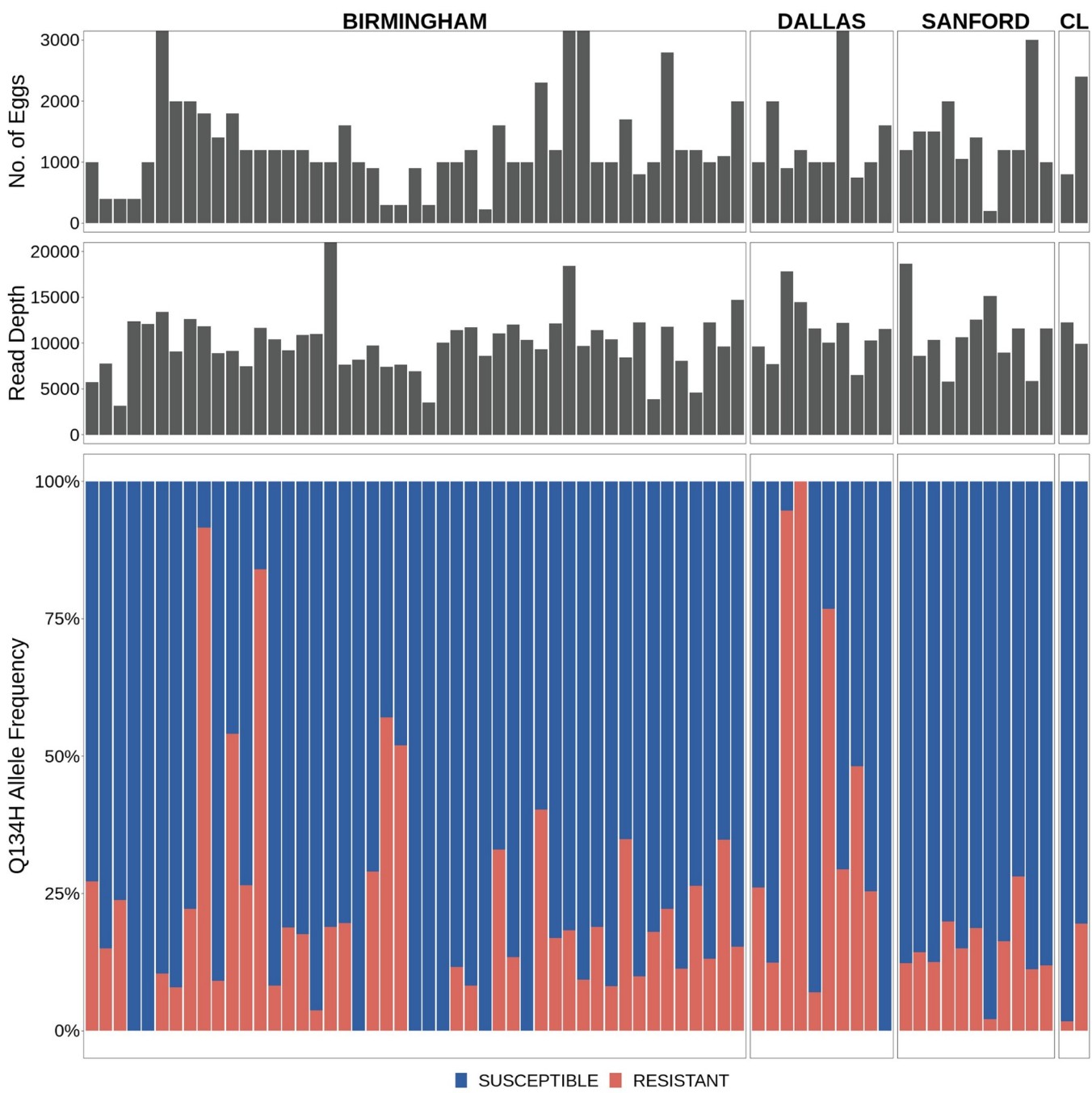

**Fig 6. Frequency and prevalence of the novel Q134H(CA<u>A</u>>CA<u>T</u>) benzimidazole resistance mutation in *A. caninum* infecting greyhounds in the USA.**
Deep amplicon sequencing data of the 293 bp isotype-1 β-tubulin fragment for *A. caninum* from greyhounds in adoption kennels in four locations in the USA [12]. The top panel is a bar chart showing the number of eggs used to make each genomic DNA preparation, in the middle is a bar chart showing the mapped read depth for each sample, and the lower chart shows the relative frequency of the Q134H(CA<u>A</u>>CA<u>T</u>) mutation. Red bars indicate the 134H(CA<u>T</u>) resistance allele and the blue bars indicate the 134Q(CAA) susceptible allele.

important benzimidazole resistance mutations in *A. caninum* allowed us to use a molecular genetic approach. We chose to use deep amplicon sequencing due to its scalability, sensitivity, and ability to provide estimates of mutation frequencies in each sample [28]. We observed that benzimidazole resistance was remarkably widespread in *A. caninum* from pet dogs across the

USA with a high prevalence of resistance mutations in 685 hookworm positive samples screened from pet dogs from various geographical locations: The F167Y(TTC>TAC) mutation was present at a prevalence of 49.7% and the Q134H(CAA>CAT) mutation at a prevalence of 31.2%, with overall mean frequencies in the positive samples of 54.0% and 16.4%, respectively. Anthelmintic resistance has not been generally considered a problem for gastrointestinal parasites of companion animals and has not been previously reported in any gastrointestinal nematode species from pet dogs or cats in the USA. Consequently, this widespread distribution of benzimidazole resistant *A. caninum* in pet dogs in the USA was not anticipated and represents a major threat to sustainable control and management of clinical cases of canine hookworm infection in the USA which is highly dependent on anthelmintic drug use. In addition, benzimidazoles and ivermectin are used to treat clinical cases of cutaneous larval migrans in humans [39] and so, management of this zoonotic condition could also be compromised.

Recent work demonstrated that multiple anthelmintic drug resistance in *A. caninum* is common in greyhounds sourced from racing and adoption kennels [14]. It seems likely that this resistance evolved as a consequence of the high transmission environment in which these animals are kept and the consequent high frequency of anthelmintic treatments that are applied, creating high drug selection pressures [11]. Our finding of widespread benzimidazole resistance in pet dogs across the USA is more surprising because pet dogs are subject to less frequent drug treatments, particularly with benzimidazoles, which are not used in pet dogs for prophylactic purposes. Consequently, benzimidazole drug selection pressure will be dramatically lower than in kenneled greyhounds. We hypothesize that the original selection for resistance occurred in greyhound kennels and the subsequent rehoming of retired greyhounds across the USA has led to the distribution of resistant *A. caninum* populations across a wide geographical range through environmental contamination [14]. Pet dogs sharing environments contaminated by drug resistant hookworm eggs will become infected, and then will be able to further spread the resistant worms. We are currently undertaking further molecular epidemiology studies to investigate this hypothesis. Although our molecular data are limited to the benzimidazole drug class, based on the recently reported data from greyhounds [14], it is likely that resistance to macrocyclic lactones and pyrantel will be similarly widespread in *A. caninum* infecting pet dogs. However, this suggestion remains to be investigated. Unfortunately, there are no molecular markers yet validated for these drug classes in *A. caninum*, so this will likely require large-scale phenotypic testing to investigate this question, at least in the short term.

The prevalence and infection intensities of *A. caninum* in pet dogs are highest in the southern USA, and generally lower in the west and the northeast regions [40,41]. This difference is likely a consequence of regional differences in ambient temperatures and humidity that favor the environmental transmission of the parasite [42,43]. An interesting result from our study is that the prevalence and frequency of both the F167Y(TTC>TAC) and Q134H(CAA>CAT) mutations are higher in *A. caninum* samples from the western USA than in the other regions. This is particularly interesting since the western region of the USA is known to have the lowest prevalence of *A. caninum* [16,40]. One possible explanation for the higher resistance prevalence and allele frequency is that there is a lower level of environmental refugia in the west due to the drier climate. Low levels of refugia lead to higher drug selection pressure and is thus recognized as an extremely important factor for the development of anthelmintic resistance in nematodes of livestock. This circumstance would therefore parallel the situation with *H. contortus* in sheep in drier and/or colder climates [44–46].

Whether Mass Drug Administration (MDA) programs will lead to selection for benzimidazole resistance in human hookworm species has been a question for many years. [47–49].

Our results in *A. caninum* have some important implications for this issue. First, the study demonstrates that isotype-1 β-tubulin mutations underlie benzimidazole resistance in a parasite that is very closely related to human hookworm species. Second, the work demonstrates the insidious nature of resistance emergence and spread: Our data conclusively show that benzimidazole resistance has become extremely widespread in canine hookworms in the USA, and this occurred with very few previous reports of resistance. Given the widespread geographical distribution and high prevalence of resistance we found, the process of spread must have been occurring for many years. The veterinary and scientific communities only recently began debating the cause of apparent hookworm treatment failures, and this only occurred after resistance to all classes of anthelmintics used in dogs had evolved [17]. This emphasizes the importance of pre-emptive surveillance if resistance is to be identified at an early enough stage for mitigation strategies to be of benefit. Thirdly, the model of intense drug treatments in kenneled greyhounds, providing the initial selection pressure for resistance emergence followed by subsequent migration and spread through the pet dog population, has some potentially important lessons for human hookworm control. Once resistance emerges regionally as a result of local drug selection, subsequent spread can occur undetected until at an advanced stage if active surveillance is not undertaken.

Finally, the results presented here provide a basis for the development and application of molecular diagnostic tests for benzimidazole resistance in canine hookworms. The data suggest that the determination of the frequency of both the F167Y(T̲T̲C>T̲A̲C) and Q134H (CA̲A̲>CA̲T̲) mutations is necessary for molecular diagnosis and surveillance of benzimidazole resistance in this parasite (Fig 1). Given the wide range in the frequency of these mutations present in different samples, diagnostic tests will need to be at least semi-quantitative to have predictive value for likely treatment success.

## The isotype-1 β-tubulin Q134H(CA<u>A</u>>CA<u>T</u>) is a novel benzimidazole resistance mutation

We previously reported that the canonical F167Y(T̲T̲C>T̲A̲C) isotype-1 β-tubulin benzimidazole resistance mutation was present in 99% (69/70) of *A. caninum* fecal egg samples from greyhounds in a number of racing and adoption kennels in the southern USA and at frequencies >50% in 62/70 samples [14]. For a set of 15 *A. caninum* populations that had phenotypic and genotypic data, we found a statistically significant correlation between the thiabendazole Egg Hatch Assay (EHA) $IC_{95}$ results and the frequency of the F167Y(T̲T̲C>T̲A̲C) mutation, which is consistent with this mutation playing an important causal role in resistance [14]. However, not all benzimidazole resistant *A. caninum* populations had a high F167Y (T̲T̲C>T̲A̲C) mutation frequency, suggesting that other β-tubulin mutation(s) could also play a role. In this study, we examined isotype-1 β-tubulin deep amplicon sequencing data available from 16 *A. caninum* populations from greyhounds shown to be benzimidazole resistant by the EHA. Thirteen of these samples had a high frequency of the F167Y(T̲T̲C>T̲A̲C) mutation consistent with their resistant phenotype. However, the remaining three samples had a low frequency of this mutation (<6%), as well as an absence of canonical resistance mutations at codons 198 and 200, suggesting that resistance could be caused by a β-tubulin mutation that has not been previously described in parasitic nematodes. Examination of the ASVs from deep sequencing identified the presence of an additional non-synonymous mutation, Q134H (CA̲A̲>CA̲T̲), at varying frequencies in 15 out of the 16 samples (Fig 1C). Notably, this novel Q134H(CA̲A̲>CA̲T̲) mutation was present at a high frequency (≥90%) in all three *A. caninum* samples (BH25, DL13, DL14) that were benzimidazole resistant but had a low frequency of the F167Y(T̲T̲C>T̲A̲C) isotype-1 β-tubulin mutation (<6%) (Fig 1B and 1C). Sequencing of the

near full-length isotype-1 β-tubulin gene revealed the Q134H(CA<u>A</u>>CA<u>T</u>) mutation to be the only non-synonymous mutation on the same haplotype. Protein-drug interaction modeling shows that the residue at codon 134 is directly in contact with the constant portion of the benzimidazole scaffold. By disrupting the contact with the wild-type glutamine, the Q134H(CA<u>A</u>>CA<u>T</u>) mutation would likely significantly reduce benzimidazole drug binding affinity (Fig 2). CRISPR-Cas9 editing in *C. elegans* showed that the Q134H(CA<u>A</u>>CA<u>C</u>) mutation is sufficient to confer a similar level of benzimidazole resistance *in vivo* as the previously described canonical resistance mutations at codons 167, 198, and 200 (Fig 3) [30]. These data indicate that Q134H (CA<u>A</u>>CA<u>T</u>) is a resistance mutation that contributes, along with the F167Y(T<u>T</u>C>T<u>A</u>C) mutation, to benzimidazole resistance in *A. caninum* in the USA greyhounds. The only previous evidence to suggest the potential of substitutions at codon 134 of a β-tubulin gene to confer benzimidazole resistance is from laboratory mutagenesis experiments on *Aspergillus nidulans* in which the Q134L mutant strain was benomyl resistant [37]. However, to our knowledge, no mutations at codon 134 have been previously identified or associated with benzimidazole resistance in natural field populations of any organism. From the extensive work that has been conducted on strongylid parasitic nematode species to date, resistance mutations have only been reported at codons 167, 198, and 200 of the β-tubulin gene, codons clustered around the drug binding site [22–25,28,50]. It is notable that a substitution at codon 134 is now added to this list of benzimidazole resistance mutations found in parasitic nematode field populations in spite of not having been identified previously in extensive studies in *C. elegans* or fungi. Work from *C. elegans* also suggests that additional resistance mutations in the isotype-1 β-tubulin gene and perhaps other loci could also be involved in different parasite species and samples [29,32,51]. These observations not only highlight the importance of directly studying target organisms of interest in the field but also the value of integrating field research on parasitic nematodes with functional analysis in a model organism, in this case, *C. elegans*.

## Materials and methods

### Collection and classification of hookworm samples from domestic dogs

From February to December 2020, hookworm positive fecal samples were collected from four different IDEXX diagnostic laboratory locations in Memphis, TN (South), North Grafton, MA (Northeast), Elmhurst, IL (Midwest), and Irvine, CA (West). Samples were refrigerated immediately after collection and stored until they were shipped to the University of Georgia. The hookworm positive samples were examined and classified into four categories based on IDEXX's semi-quantitative classification: "rare": 1–2 ova per slide, "few": 3–10 ova per slide, "moderate": 11–30 ova per slide, and "many": >30 ova per slide.

The samples were assigned to four geographic regions—West, Midwest, South, and Northeast [39,52]. The states comprising each region were as follows: West (AK, AZ, CA, CO, HI, ID, MT, NM, NV, OR, UT, WA, WY), Midwest (IA, IL, IN, KS, MI, MN, MO, ND, NE, OH, SD, WI), South (AL, AR, FL, GA, KY, LA, MS, NC, OK, SC, TN, TX, VA, WV), and Northeast (CT, DE, MA, MD, ME, NH, NJ, NY, PA, RI, VT).

The dogs from which the samples were collected were categorized into small, medium, and large breeds according to the American Kennel Club (AKC, 2021) and into age groups as per recent American Animal Hospital Association guidelines: puppy: less than 1 year of age, young adult: 1 to 3 years, mature adult: 4 to 6 years, and senior: greater than 6 years old [53].

### Fecal floatation and DNA extraction

Eggs from hookworm positive fecal samples classified as "many", "moderate", and "few" were isolated individually. Samples classified as "rare", as well as those classified as "few" but with

<1 g of feces available, were combined into pools before isolating the eggs. Wooden tongue depressors were used to mix the fecal samples with water to produce a fecal slurry, which was then filtered through a bleached grade 40 cheesecloth (GRAINGER, Lake Forest, IL) and transferred to disposable cups. The filtered fecal solution was then transferred to a 15 mL falcon tube and centrifuged at 240 x *g* for 10 minutes. The supernatant was discarded and 10 mL of sodium nitrate (Feca-Med, Vedco, Inc. St. Joseph; MO, USA specific gravity = 1.25 to 1.30) was added, followed by vortexing to break up the pellet, which was centrifuged again at 240 x *g* for 10 minutes. Following centrifugation, hookworm eggs were recovered from the supernatant, rinsed with distilled water through a 20 μM sieve, and transferred to a tube. Three aliquots of 20 μL each were used to estimate the number of eggs. Following estimation, the eggs were transferred to 2 mL cryotubes (Sigma-Aldrich, St. Louis, MO), suspended in a final concentration of 70% ethanol, and stored at -80˚C until further use. DNA extraction was performed on the individual and pooled egg samples as previously described with minor modifications [14]. Briefly, three freeze-thaw cycles were carried out at -80˚C and at 55˚C respectively, followed by the addition of 180 μL of DirectPCR (Cell) Lysis Buffer (Catalog No. 301-C, Viagen Biotech, St. Louis, MO) and 20 μL of Proteinase K (Catalog No. 19133, QIAGEN, Hilden, Germany). Samples were then incubated for at least 12 hours at 65˚C, followed by a one-hour incubation at 95˚C, and were then cooled to 4˚C. DNA was purified from the crude DNA lysates using the QIAGEN QIAmp DNA mini kit (Catalog No. 51306) following the manufacturer's recommended protocol. The purified DNA was eluted in 50 μL of Tris-EDTA buffer solution (Sigma-Aldrich, St. Louis, MO) and stored at -80˚C.

## Sequencing the full-length isotype-1 β-tubulin from *Ancylostoma caninum*

Primer pairs were designed to amplify the 3,467 bp near full-length sequence of the isotype-1 β-tubulin gene in *A. caninum*. The forward and reverse primer sequences are as follows: AC_BT1_Fwd: 5'-GTGAGATCGTGCATGTACAAGC-3' and AC_BT1_Rev: 5'-ATGGGTGATCTCGA TGCGGA-3' (S1A Fig). The near full-length isotype-1 β-tubulin gene was amplified from hookworm egg genomic DNA preparations from two greyhound samples, BH31 and BH45, that contained the Q134H(CA<u>A</u>>CAT) mutation at 84.0% and 44.4%, respectively, and also from a susceptible *A. caninum* sample (Barrow) [17]. The aim was to sequence the haplotype of the near full-length isotype-1 β-tubulin gene to determine if any additional non-synonymous mutations were present. The following PCR conditions were used: 12.5 μL of repliQa HiFi ToughMix (QuantaBio, USA), 1 μL of 10 μM Forward primer, 1 μL of 10 μM Reverse primer, 8.5 μL of molecular grade water, and 2 μL of DNA template. The thermocycling conditions were 98˚C for 3 minutes, followed by 40 cycles of 98˚C for 10 seconds, 68˚C for 5 seconds, and 68˚C for 30 seconds.

PCR amplicons were purified using the QIAGEN QIAquick PCR Purification kit (Catalog No. 28104) and cloned using the TOPO XL-2 Complete PCR Cloning Kit (ThermoFisher Scientific, USA) with One Shot TOP10 Chemically Competent *E. coli* (Invitrogen, USA). Plasmid DNA was prepared using the QIAGEN QIAprep Spin Miniprep kit (Catalog No. 27106X4) and sequenced using the Applied Biosystems 3730xl (96 capillary) Genetic Analyzer using Big-Dye Terminator chemistry (Cumming School of Medicine, University of Calgary). Three, two, and one clones were sequenced from BH45, BH31, and Barrow, respectively, using primers for Sanger sequencing (S2 Table).

## CRISPR-Cas9 editing of the Q134H(CAA>CAT) allele into the *C. elegans* *ben-1* gene

*C. elegans* were cultured on plates of modified nematode growth media (NGMA), containing 1% agar and 0.7% agarose, seeded with OP50 bacteria [54]. Plates were maintained at 20˚C for

the duration of this experiment. Before the assay, animals were grown for three generations to reduce the multigenerational effects of starvation [55]. The wild-type laboratory reference strain N2 and the previously created *ben-1(ean64)* deletion strain (ECA882) [29] were used to verify quantitative sensitivity and resistance, respectively. Two strains containing the Q134H (CA<u>A</u>>CA<u>C</u>) allele were generated in the N2 background using CRISPR-Cas9 genome editing as previously described [29,30]. Briefly, after injection, possibly edited strains underwent multiple generations of confirmation, ensuring that strains were homozygous for the Q134H (CA<u>A</u>>CA<u>C</u>) allele. Two independently edited strains, *ben-1(ean243)* and *ben-1(ean244)*, were created to control for any potential off-target effects.

A previously described high-throughput fitness assay was used for all measurements of benzimidazole responses [29–31]. The assay included growth measurements of 44 replicate populations comprising approximately 50 animals for each strain in each condition, with the exception of N2 in the albendazole condition (Data Available at: https://github.com/AndersenLab/2022_BZ_Resistance_ben1_Q134H).Each strain was amplified on NGMA plates for four generations, bleach synchronized, and then approximately 50 embryos in 50 μL of K medium [56] were aliquoted into each well of a 96-well plate. The following morning, arrested L1s were given a mixture of HB101 bacterial lysate [57] at a concentration of 5 mg/mL, kanamycin at a concentration of 50 μM, and either 1% DMSO or 30 μM albendazole in 1% DMSO. Animals were grown for 48 hours with constant shaking. After growing, animals were treated with 50 mM sodium azide in M9 buffer to straighten the animals for scoring [58]. Each individual was measured using the COPAS BIOSORT large particle flow cytometer (Union Biometrica, Holliston MA). Animal optical density normalized by animal length (median.EXT) was calculated for each individual. Optical density is reduced in susceptible animals after exposure to benzimidazoles and is able to differentiate resistant and susceptible animals [30,31]. Median optical density of populations in each well was used for quantification of albendazole response. Higher regressed optical density values indicate reduced anthelmintic response, and lower values indicate a more severe response to anthelmintic exposure.

For the analysis of optical density, all data processing was performed using the R package *easysorter* [59] as described previously [29,30]. Contaminated wells and outliers were removed, and the data were regressed using the *regress()* function in the *easysorter* package to remove control phenotypes from the experimental phenotypes, leaving residual values for analysis. Statistical comparisons were performed in R (Version 4.1.2). Tukey's Honest Significant Difference (HSD) test was performed using the *rstatix* package [60] on an ANOVA model with the formula "phenotype~strain" to calculate differences among population phenotypes.

## Predictive structural modeling of the isotype-1 β-tubulin protein—benzimidazole drug interactions

The structural model for the *Ancylostoma caninum* isotype-1 β-tubulin was made using Alpha-Fold2 [61] and was then aligned to Porcine β-tubulin bound to nocodazole (PDB 5CA1) using Pymol Version 2.4 [62]. The canonical as well as the novel isotype-1 β-tubulin codons associated with benzimidazole resistance (134, 167, 198, and 200) were inspected and are in direct van der Waals contact with the constant portion of benzimidazole drugs.

## Deep amplicon sequencing and analysis

Deep amplicon sequencing was used to identify and determine the frequency of sequence polymorphisms at codons 134, 167, 198, and 200 of the *A. caninum* isotype-1 β-tubulin as previously described [14,17,28]. Two separate fragments, a 293 bp fragment encompassing codons 134 and 167, and a 340 bp fragment, encompassing codons 198 and 200, were PCR-amplified as previously

described [14]. Sample purification, indexing, and library preparation for Illumina deep sequencing were performed as previously described [28]. Libraries were sequenced on the Illumina MiSeq platform using the 2 x 250 bp V2 sequencing chemistry (Illumina Inc., USA). One positive control and eight negative controls comprising a benzimidazole-susceptible *A. caninum* fecal egg sample and molecular-grade water, respectively, were included on each 96-well plate.

Following sequencing, the paired-end reads were analyzed using the previously described bioinformatics pipeline with slight modifications [14,63]. As per the developer's recommendation, the reads from each run were processed separately using the DADA2 pipeline, to reduce the error rates. The following parameters were used in addition to the default settings during the quality filtering step in the DADA2 pipeline: i) based on read quality, the forward and reverse reads were trimmed to a length of 220 bp and 150 bp, respectively, for the fragment containing codons 134 and 167; whereas the forward and reverse reads for the fragment containing codons 198 and 200 were trimmed to 220 bp and 200 bp, respectively; ii) reads shorter than 50 bp or with an expected error of >1 and >2 in the forward and reverse reads, respectively, were removed. The filtered and merged reads from each run were then combined, the chimeric sequences were removed using DADA2 and variant analysis was performed as previously described [14]. Variant calling was performed by aligning the generated ASVs to the *A. caninum* isotype-1 β-tubulin reference sequence (Genbank Accession: DQ459314.1) using a global (Needleman-Wunsch) pairwise alignment algorithm without end gap penalties. Following alignment, the ASVs were discarded if they were <180 bp or >350 bp long, or if they had a percentage identity <70% to the reference sequence, or if the ASVs had fewer than 200 reads in a sample, or if they were not present in two or more samples. This additional filtering ensures the removal of spurious sequences. The filtered reads from each step for the two amplicons are given in S3 and S4 Tables. Additionally, we excluded all samples with a read depth lower than 1000 reads from further analysis. Results of the variant analysis were visualized in R (Version 4.1.0) using the packages *ggplot2* and *leaflet* [64,65].

### Assessment of the repeatability of SNP frequencies determined by amplicon sequencing

In order to test the repeatability of the benzimidazole resistance SNP frequencies determined by amplicon sequencing, two independent PCRs were performed on 50 randomly chosen samples and sequenced on two separate Illumina Miseq runs. Lin's Concordance Correlation Coefficient was used to calculate the agreement between the resistance polymorphism frequencies for these 50 replicates. All statistical analyses were performed in R (Version 4.1.0). A high level of agreement was observed in the frequency of the F167Y(T<u>T</u>C>T<u>A</u>C) resistance mutation between the replicates (Lin's Concordance Correlation Coefficient CCC = 0.94; 95% C.I. = 0.90, 0.97) (S6A Fig). The level of agreement between the Q134H(CA<u>A</u>>CA<u>T</u>) frequencies in the replicates was moderate (Lin's Concordance Correlation Coefficient CCC = 0.62; 95% C.I. = 0.43, 0.75) (S6B Fig).

### Statistical analysis

Differences in overall and pairwise comparisons of resistance mutation prevalences between the regions, age groups, and breed sizes were calculated by Pearson's Chi-squared and Fisher's test with Bonferroni correction, respectively. Differences in the overall and pairwise comparisons of the resistance mutation frequencies among the regions were calculated by Kruskal-Wallis rank-sum test and Wilcoxon's rank-sum test with Bonferroni correction, respectively. Bootstrap 95% confidence intervals for the mean resistance allele frequencies were calculated using the *boot* package in R [66]. All statistical analyses were performed in R (Version 4.1.0) using the *rstatix* package [60].

## Supporting information

**S1 Appendix.** *A. caninum* **pooled samples across the USA.**
(DOCX)

**S1 Fig. Full-length genomic sequences of the *A. caninum* isotype-1 β-tubulin gene showing the sequence polymorphisms present.** (A): Multiple Sequence Alignment of the 3,467 bp long *A. caninum* isotype-1 β-tubulin near full-length genomic sequence cloned and sequenced from greyhound A. caninum samples BH31 and BH45, the benzimidazole-susceptible lab sample (Barrow), and the *A. caninum* isotype-1 β-tubulin reference sequence (Genbank accession: DQ459314). Following Sanger sequencing, the sequences of each clone were assembled *de novo* using the Geneious software and the assembled consensus sequences for each clone were aligned using the MUSCLE tool for multiple sequence alignment in Geneious v. 10.0.9. The synonymous and non-synonymous polymorphisms relative to the reference sequence, DQ459314, are highlighted in black and red, respectively. (B): Distance matrix showing the pairwise number of nucleotide differences among the sequences.
(TIF)

**S2 Fig. *In-silico* protein structural model of the *A. caninum* isotype-1 β-tubulin bound to nocodazole with modeled thiabendazole.** Structural model for *A. caninum* isotype-1 β-tubulin made using AlphaFold2 (PMID: 34265844) (purple) aligned to Porcine β-tubulin (not shown) bound to nocodazole (white) (from PDB 5CA1) with thiabendazole modeled (brown). Potential interaction with residue 198 shown (dashed lines) with other features occupying the similar volume in the pocket to the methyl ester terminus.
(TIF)

**S3 Fig. Frequency of the 167Y(TAC) resistance mutation in *A. caninum* from dogs of different breeds and age groups.** (A) Violin Plot of the 167Y(TAC) resistance allele frequencies in *A. caninum* from dogs of different breed sizes. The mean frequency is indicated by a horizontal line and any statistically significant differences calculated using the pairwise Wilcoxon rank sum test ($p<0.05$) between the regions are indicated (p-value $>0.05$ not indicated). (B) Violin Plot of the 167Y(TAC) resistance allele frequencies in *A. caninum* from dogs of different age groups. The mean frequency is indicated by a horizontal line and any statistically significant differences calculated using the pairwise Wilcoxon rank sum test ($p<0.05$) between the regions are indicated (p-value $>0.05$ not indicated).
(TIF)

**S4 Fig. Absence of resistance mutations at codons 198 and 200 of the *A. caninum* isotype-1 β-tubulin gene.** Deep amplicon sequencing data of the 340 bp fragment of *A. caninum* from the 312/ 328 individual samples across the USA. The top panel is a bar chart showing the number of eggs used to make each genomic DNA preparation, in the middle is a bar chart showing the mapped read depth for each sample, and the lower charts show the frequency of resistance or susceptible alleles present at codons 198 and 200 of the isotype-1 β-tubulin gene. Blue bars indicate the susceptible alleles 198E(GAA) and 200F(TTC) at both the codons.
(TIF)

**S5 Fig. Frequency of the 134H(CAT) resistance mutation in *A. caninum* from dogs of different breeds and age groups.** (A) Violin Plot of the 134H(CAT) resistance allele frequencies in *A. caninum* from dogs of different breed sizes. The mean frequency is indicated by a horizontal line and any statistically significant differences calculated using the pairwise Wilcoxon rank sum test ($p<0.05$) between the regions are indicated (p-value $>0.05$ not indicated). (B) Violin Plot of the 134H(CAT) resistance allele frequencies in *A. caninum* from dogs of

different age groups. The mean frequency is indicated by a horizontal line and any statistically significant differences calculated using the pairwise Wilcoxon rank sum test (p<0.05) between the regions are indicated (p-value >0.05 not indicated).
(TIF)

**S6 Fig. Assessment of the repeatability of SNP frequencies determined by amplicon sequencing.** (A): Scatterplot comparing the 167Y(T<u>A</u>C) resistance mutation frequencies determined from two independent replicate PCRs and Miseq runs for 50 samples. 24/50 samples carried the resistance mutation. The solid diagonal line indicates perfect agreement between the two runs. Lin's Concordance Correlation Coefficient (CCC) indicates the level of agreement between the two sets of replicates. (B): Scatterplot comparing the 134H(CA<u>T</u>) resistance mutation frequencies determined from two independent replicate PCRs and Miseq runs for 50 samples. 10/50 samples carried the resistance mutation. The solid diagonal line indicates perfect agreement between the two runs. Lin's Concordance Correlation Coefficient (CCC) indicates the level of agreement between the two sets of replicates.
(TIF)

**S7 Fig. Frequency of the benzimidazole resistance mutations in pooled samples of *A. caninum* across the USA.** (A): Deep amplicon sequencing data of the 293 bp isotype-1 β-tubulin fragment of *A. caninum* from the 63/ 65 pooled samples across the USA. The top panel is a bar chart showing the number of eggs used to make each genomic DNA preparation, in the middle is a bar chart showing the mapped read depth for each sample, and the lower chart shows the relative frequency of the F167Y(T<u>T</u>C>T<u>A</u>C) mutation. Red bars indicate the 167Y(T<u>A</u>C) resistance allele and the blue bars indicate the 167F(TTC) susceptible allele. (B): Deep amplicon sequencing data of the 293 bp isotype-1 β-tubulin fragment of *A. caninum* from the 63/ 65 pooled samples across the USA. The top panel is a bar chart showing the number of eggs used to make each genomic DNA preparation, in the middle is a bar chart showing the mapped read depth for each sample, and the lower chart shows the relative frequency of the Q134H (CA<u>A</u>>CA<u>T</u>) mutation. Red bars indicate the 134H(CA<u>T</u>) resistance allele and the blue bars indicate the 134Q(CAA) susceptible allele.
(TIF)

**S8 Fig. Absence of resistance mutations at codons 198 and 200 in the isotype-1 β-tubulin of A. caninum from pooled samples.** Deep amplicon sequencing data of the 340 bp fragment of A. caninum from the 63 pooled samples across the USA. The top panel is a bar chart showing the number of eggs used to make each genomic DNA preparation, in the middle is a bar chart showing the mapped read depth for each sample, and the lower charts show the frequency of resistance or susceptible alleles present at codons 198 and 200 of the isotype-1 β-tubulin gene. Blue bars indicate the susceptible alleles 198E(GAA) and 200F(TTC) at both the codons.
(TIF)

**S1 Table. Information on pooled samples of *A. caninum*.** The table contains information on the 65 pooled samples that were used in the study, their geographical region, the number of samples in each pool, and the number of eggs that were used for genomic DNA preparation.
(DOCX)

**S2 Table. Primers used for Sanger sequencing of the near full-length isotype-1 β-tubulin.**
(DOCX)

**S3 Table. Sequenced reads filtering for the 293 bp fragment encompassing codons 134 and 167.** The table contains information on the number of sequenced reads for each sample, the

number of reads that were filtered during each step of the DADA2 pipeline and the variant calling pipeline for the 293 bp fragment.
(XLSX)

**S4 Table. Sequenced reads filtering for the 340 bp fragment encompassing codons 198 and 200.** The table contains information on the number of sequenced reads for each sample, the number of reads that were filtered during each step of the DADA2 pipeline and the variant calling pipeline for the 340 bp fragment.
(XLSX)

## Acknowledgments

We thank the people at IDEXX who helped provide us with the hookworm samples: David Elsemore, Troy Goddu, Jancy Hanscom, Rebecca Foley, and Molly Donovan, as well as the rest of the staff at the different IDEXX laboratories.

## Author Contributions

**Conceptualization:** Abhinaya Venkatesan, Pablo D. Jimenez Castro, Ray M. Kaplan, John S. Gilleard.

**Data curation:** Abhinaya Venkatesan.

**Formal analysis:** Abhinaya Venkatesan.

**Funding acquisition:** Erik C. Andersen, Ray M. Kaplan, John S. Gilleard.

**Investigation:** Abhinaya Venkatesan, Arianna Morosetti, Hannah Horvath, Elizabeth Redman, James Bryant Collins, James S. Fraser.

**Methodology:** Abhinaya Venkatesan.

**Project administration:** Abhinaya Venkatesan, Pablo D. Jimenez Castro, Erik C. Andersen, Ray M. Kaplan, John S. Gilleard.

**Resources:** Pablo D. Jimenez Castro, Kayla Dunn, Ray M. Kaplan, John S. Gilleard.

**Software:** Abhinaya Venkatesan, Rebecca Chen.

**Supervision:** James S. Fraser, Erik C. Andersen, Ray M. Kaplan, John S. Gilleard.

**Validation:** Abhinaya Venkatesan, Arianna Morosetti, Hannah Horvath, James Bryant Collins.

**Visualization:** Abhinaya Venkatesan, Rebecca Chen, James Bryant Collins, James S. Fraser.

**Writing – original draft:** Abhinaya Venkatesan, John S. Gilleard.

**Writing – review & editing:** Abhinaya Venkatesan, Pablo D. Jimenez Castro, James S. Fraser, Erik C. Andersen, Ray M. Kaplan, John S. Gilleard.

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
