## [Decision Letter · Decision Letter 0]

22 Nov 2022

Dear Ms Venkatesan,

Thank you very much for submitting your manuscript "Molecular evidence of widespread benzimidazole drug resistance in Ancylostoma caninum from domestic dogs throughout the USA and discovery of a novel β-tubulin benzimidazole resistance mutation" for consideration at PLOS Pathogens. As with all papers reviewed by the journal, your manuscript was reviewed by members of the editorial board and by several independent reviewers. In light of the reviews (below this email), we would like to invite the resubmission of a significantly-revised version that takes into account the reviewers' comments.

Three reviewers have given their thoughts and I have read the manuscript as well. Two of the three reviewers (#1,3) believe the manuscript require edits, with reviewer 1 providing an extensive list of points for clarification. Reviewer 2 believes the manuscript is not of sufficient significance for PLoS Pathogens. Whereas I agree more with the opinion of reviewers 1 and 3 and believe the results are important with merit, I do believe that the paper would benefit by addressing the criticism of reviewer 2 and modification of the manuscript. In addition, the points raised by reviewer 1 are very similar to my own thoughts (eg, areas where I find the presentation confusing) and all of these should be addressed.

We cannot make any decision about publication until we have seen the revised manuscript and your response to the reviewers' comments. Your revised manuscript is also likely to be sent to reviewers for further evaluation.

Sincerely,

Raffi V. Aroian

Guest Editor

PLOS Pathogens

P'ng Loke

Section Editor

PLOS Pathogens

Kasturi Haldar

Editor-in-Chief

PLOS Pathogens

orcid.org/0000-0001-5065-158X

Michael Malim

Editor-in-Chief

PLOS Pathogens

orcid.org/0000-0002-7699-2064

Three reviewers have given their thoughts and I have read the manuscript as well. Two of the three reviewers (#1,3) believe the manuscript require edits, with reviewer 1 providing an extensive list of points for clarification. Reviewer 2 believes the manuscript is not of sufficient significance for PLoS Pathogens. Whereas I agree more with the opinion of reviewers 1 and 3 and believe the results are important with merit, I do believe that the paper would benefit by addressing the criticism of reviewer 2 and modification of the manuscript. In addition, the points raised by reviewer 1 are very similar to my own thoughts (eg, areas where I find the presentation confusing) and all of these should be addressed.

Reviewer's Responses to Questions

**Part I - Summary**

Reviewer #1: Venkastesan et al: Molecular evidence of widespread benzimidazole drug resistance in Ancylostoma caninum from domestic dogs throughout the USA and discovery of a novel β-tubulin benzimidazole resistance mutation.

The manuscript describes observations on benzimidazole anthelmintic resistant Ancylostoma caninum associated with point mutations in the β-tubulin gene and identify a previously un-recognized site, Q134. The observations are original, significant and are anticipated to have an impact on the field of study. The authors should address the critiques before publication.

Reviewer #2: This manuscript, entitled “Molecular evidence of widespread benzimidazole drug resistance in Ancylostoma caninum from domestic dogs throughout the USA and discovery of a novel β-tubulin benzimidazole resistance mutation" (PPATHOGENS-D-22-01783) examined pet dog hookworm populations from throughout the US for the presence of a know β-tubulin mutation (F167Y) that confers resistance to benzimidazole drugs. The authors used state of the art methodology to determine allele frequencies and showed that the allele has spread widely within the pet dog population. Perhaps more importantly, the authors identified a novel mutation (Q134H) associated with resistance and demonstrated that the homologous mutation introduced into C. elegans confers BZ resistance. While the data is robust, it is descriptive in nature, and some of the conclusions are unwarranted. Specifically:

1. Proclamation that this represents the first cases of drug resistant hookworms from pets is not accurate. Members of this group reported a miniature schnauzer (Tara) that was resistant to BZ as well as other anthelmintics (Jimenez Castro et al, 2019). Furthermore, while other reports of multidrug resistant hookworms were from greyhounds (Kitchen et al, 2019; Jimenez Castro et al 2019, 2021), these were rescued animals and considered pets. It was already clear that multidrug resistant hookworms have been slowly spreading into the pet population, and therefore it is disingenuous to claim this as the first report from pets.

2. Furthermore, it is a stretch to claim that drug resistance is “widespread” in the US based on the frequency of the F167Y allele alone. Given that phenotypic resistance requires the allele to be homozygous and is generally not detectable until it reaches at least 25% of the population, there are few places outside the West that have sufficiently high frequencies to conclude clinical resistance is present. What does seem apparent is that there are several “hotspots” where resistant hookworms are likely to be common, such as CA, IL and New England. The author should tone down the claims of resistance, and instead use phrases like an increased likelihood of resistance or high frequencies of resistance alleles.

Reviewer #3: I find exceptional merit in this manuscript. It has highly significan relevance for the potential of developing drug resistance in human hookworm populations as MDA campaigns intensify. The parasite under study is a zoonotic parasite (cutaneous larva migrans), which alone renders it suitable for this journal. The manuscript is well-written and concise. it blends an appealing mix of methods and the experiments lead to solid conclusions. The experimental design is sound, the figures clear and the conclusions fully justified. I can only congratulate the authors on their achievement.

**Part II – Major Issues: Key Experiments Required for Acceptance**

Reviewer #1: (No Response)

Reviewer #2: None

Reviewer #3: No new experiments are required.

**Part III – Minor Issues: Editorial and Data Presentation Modifications**

Reviewer #1: Generally

The benzimidazole anthelmintics, although similar, do have differences in their chemical structure but are assumed to be homogenous without showing any diversity for binding to different mutants of the isotype-1β-tubulin gene. Examples are:

1) The benzimidazole anthelmintics that were used for the treatment of the dogs are not specified;

2) Thiabendazole is used for the Egg Hatch Assay (page 14);

3) Albendazole, which is not used in dogs, is used to assess resistance in the ean243 and ean244; nocodazole, which is an antineoplastic agent and not used as an anthelmintic is used for structural modeling of the A. caninum isotype-1 β-tubulin.

4) The absence of the canonical codon 198 and 200 benzimidazole resistance mutations may relate to the use absence of benzimidazole anthelmintics used for large animals.

Abstract

1. Briefly, define prevalences and overall frequency. The use of frequency is sometimes confusing in the manuscript.

Introduction

1. 2nd paragraph: Comment on the prior use of vaccination against Ancylostoma caninum using irradiated l3 larvae e.g. Miller 1965, J Parasitology.

2. 3rd paragraph: Comment on any zoonotic concerns with the increase in observations from dog parks.

3. 4th paragraph: Comment on benzimidazole resistance in fungi and C. elegans that has been associated with other amino acid mutations, even though it is commented on later in the discussion. Some examples include: A165, H6, E198 and F200 Minagawa, Cells 2021 and; C. elegans A185P, E69G, Q131L, S145F, M257L and D404N variants (Hahnel et al. 2018).

Results

1. Page 7. Top: Were any other mutations associated with resistance in fungi or C. elegans found at even low frequencies with the deep amplicon sequencing?

2. Page 7, 2nd paragraph: Mapping with nocodazole, contact with methyl ester terminus relevant for albendazole, mebendazole, and fenbendazole but not thiabendazole that has a thiazole group present. How does the thiazole group of thiabendazole fit in the model? This is clinically relevant when it is used to diagnose benzimidazole resistance with the EHAs.

3. Page 8, Top: Albendazole responses to the ean243 and ean244 suggest resistance. Do they show the same resistance to thiabendazole as well because thiabendazole is used in the EHAs (page 14) and the benzimidazole anthelmintics used for dogs?

Discussion

1. Briefly comment if there is knowledge of dominance, recessive, homozygous, heterozygous and sex-linked effects of benzimidazole resistance and how this would affect spread.

2. Page 14: Not clear: ‘ …present in 99% (69/70) of A. caninum isolates sampled from greyhound……..and at high frequencies in most cases (> 50% in 62/70 isolates)…

Figures

Fig 1. Log IC95 values. Are the values -Log µM concentrations? What is the resistance threshold in µM?

Label Frequency as % ?

Fig 2. Thiabendazole Egg Hatch Assays were used for resistance estimations. More appropriate to fit thiabendazole than nocodazole which is an antineoplastic agent rather than an anthelmintic.

Reviewer #2: I did not find anything significant.

Reviewer #3: I have only two minor concerns. First, the last paragraph of the Introduction presents the results and conclusions of the work; it belongs, if anywhere, in the Discussion. Second, the authors should stress the zoonotic aspect of this parasite; while cases of CLM in the USA are rare, the increasing incidence of infection in companion animals and the inability to treat this MDR population is a public health threat and should be discussed. Finally, it behooves the authors to at least briefly discuss treatment options for MDR A. caninum in dogs and how to best diagnose this particular infection at this time. It is important to stress that the two main treatments for human CLM cases, albendazole and ivermectin, may be ineffective going forward.

PLOS authors have the option to publish the peer review history of their article (what does this mean?). If published, this will include your full peer review and any attached files.

Reviewer #1: No

Reviewer #2: No

Reviewer #3: No
---

## [Decision Letter · Decision Letter 1]

22 Jan 2023

Dear Ms Venkatesan,

We are pleased to inform you that your manuscript 'Molecular evidence of widespread benzimidazole drug resistance in Ancylostoma caninum from domestic dogs throughout the USA and discovery of a novel β-tubulin benzimidazole resistance mutation' has been provisionally accepted for publication in PLOS Pathogens.

Best regards,

Raffi V. Aroian

Guest Editor

PLOS Pathogens

P'ng Loke

Section Editor

PLOS Pathogens

Kasturi Haldar

Editor-in-Chief

PLOS Pathogens

orcid.org/0000-0001-5065-158X

Michael Malim

Editor-in-Chief

PLOS Pathogens

orcid.org/0000-0002-7699-2064

Reviewer Comments (if any, and for reference):

Reviewer's Responses to Questions

**Part I - Summary**

Reviewer #1: As before this referee is of the opinion that the manuscript is well executed, is significant and will have an impact on the field of study.

Reviewer #3: The revisions made in response to my minor concerns and those of the other reviewers have fully resolved them. No further changes are suggested.

**Part II – Major Issues: Key Experiments Required for Acceptance**

Reviewer #1: No major issues

Reviewer #3: (No Response)

**Part III – Minor Issues: Editorial and Data Presentation Modifications**

Reviewer #1: The authors have responded positively and appropriately to the different reviewers' critiques

Reviewer #3: (No Response)

PLOS authors have the option to publish the peer review history of their article (what does this mean?). If published, this will include your full peer review and any attached files.

Reviewer #1: No

Reviewer #3: No

---

## [Editor Report · Acceptance letter]

27 Feb 2023

Dear Prof. Gilleard,

We are delighted to inform you that your manuscript, "Molecular evidence of widespread benzimidazole drug resistance in Ancylostoma caninum from domestic dogs throughout the USA and discovery of a novel β-tubulin benzimidazole resistance mutation," has been formally accepted for publication in PLOS Pathogens.

Best regards,

Kasturi Haldar

Editor-in-Chief

PLOS Pathogens

orcid.org/0000-0001-5065-158X

Michael Malim

Editor-in-Chief

PLOS Pathogens

orcid.org/0000-0002-7699-2064